# Targeted Redox Regulation α-Ketoglutarate Dehydrogenase Complex for the Treatment of Human Diseases

**DOI:** 10.3390/cells14090653

**Published:** 2025-04-29

**Authors:** Ryan J. Mailloux

**Affiliations:** School of Human Nutrition, McGill University, 21111 Lakeshore Road, Sainte-Anne-de-Bellevue, Quebec, QC H9X 3V9, Canada; ryan.mailloux@mcgill.ca

**Keywords:** KGDHc, oxidative eustress, oxidative distress, succinate, NAFLD, mitochondria, hydrogen peroxide, metabolic diseases

## Abstract

α-ketoglutarate dehydrogenase complex (KGDHc) is a crucial enzyme in the tricarboxylic acid (TCA) cycle that intersects monosaccharides, amino acids, and fatty acid catabolism with oxidative phosphorylation (OxPhos). A key feature of KGDHc is its ability to sense changes in the redox environment through the reversible oxidation of the vicinal lipoic acid thiols of its dihydrolipoamide succinyltransferase (DLST; E2) subunit, which controls its activity and, by extension, OxPhos. This characteristic inculcates KGDHc with redox regulatory properties for the modulation of metabolism and mediating of intra- and intercellular signals. The innate capacity of KGDHc to participate in the regulation of cell redox homeodynamics also occurs through the production of mitochondrial hydrogen peroxide (mtH_2_O_2_), which is generated by the dihydrolipoamide dehydrogenase (DLD; E3) downstream from the E2 subunit. Reversible covalent redox modification of the E2 subunit controls this mtH_2_O_2_ production by KGDHc, which not only protects from oxidative distress but also modulates oxidative eustress pathways. The importance of KGDHc in modulating redox homeodynamics is underscored by the pathogenesis of neurological and metabolic disorders that occur due to the hyper-generation of mtH_2_O_2_ by this enzyme complex. This also implies that the targeted redox modification of the E2 subunit could be a potential therapeutic strategy for limiting the oxidative distress triggered by KGDHc mtH_2_O_2_ hyper-generation. In this short article, I will discuss recent findings demonstrating KGDHc is a potent mtH_2_O_2_ source that can trigger the manifestation of several neurological and metabolic diseases, including non-alcoholic fatty liver disease (NAFLD), inflammation, and cancer, and the targeted redox modification of the E2 subunit could alleviate these syndromes.

## 1. Introduction

α-ketoglutarate dehydrogenase complex (KGDHc) is the fourth enzyme in the TCA cycle and connects the metabolism of monosaccharides, fatty acids, and amino acids with OxPhos (Figure 1). It is a large ~3.2 MDa multi-subunit enzyme complex that is composed of ~12 E1 (α-ketoglutarate decarboxylase) and ~12 E3 subunits surrounding a 24-mer E2 core (Figure 1) [1]. Recent evidence shows the multimer structure of KGDHc is dynamic, with the E2 core connected to surrounding E1 and E3 subunits through highly flexible linker regions connected to lipoyl domains [2]. Like the other enzymes in the α-keto acid dehydrogenase complex (KDHc) family, the activity of KGDHc depends on the E1, E2, and E3 subunits arranged in a 1:1:1 ratio [3,4,5,6,7] (Figure 1). A variety of cofactors and prosthetic groups are required for KGDH to catalyze the multi-step reaction that couples the oxidative decarboxylation of α-ketoglutarate to the generation of succinyl-CoA and NADH. Co-factors and prosthetic groups include thiamine pyrophosphate (E1), CoASH, and lipoic acid, which is covalently attached to lysine (E2), and NAD^+^ and FAD (E3) [3,4,5,6,7]. The E1 subunit drives the decarboxylation of α-ketoglutarate, forming a succinyl-TPP intermediate [3,4,5,6,7]. The acyltransferase activity of the E2 then catalyzes a thiol disulfide exchange reaction between lipoate, CoASH, and succinyl-TPP, producing succinyl-CoA and dihydrolipoamide [3,4,5,6,7]. Reducing equivalents in the dihydrolipoamide are then transferred to the FAD in the E3 subunit through a CxxxxC redox motif, generating FADH_2,_ which then transfers a hydride to NAD^+^, forming NADH [3,4,5,6,7,8]. This multi-step reaction is thermodynamically irreversible, although the E3 subunit can oxidize NADH, which results in mitochondrial superoxide (mtO_2_^•−^)/hydrogen peroxide (mtH_2_O_2_) production by the E3 subunit (discussed below).

The catalytic features of KGDHc and its importance in facilitating the metabolism of various mitochondrial fuel sources make the enzyme complex an ideal site for metabolic regulation. KGDHc is controlled by several allosteric regulators, including [NADH]/[NAD^+^], [succinyl-CoA]/[CoASH], ATP, ADP, and cations like calcium [3,4,5,6,7]. KGD4, which was originally identified as a component of the mitochondrial ribosome, was proposed to be an important molecular adaptor for the assembly of KGDHc [9]. This led to speculation that it may play a role in the modulation of KGDHc activity [10]. However, recent work found no evidence that KGD4 is required to facilitate the inter-subunit interactions needed to form the multimer complex of KGDHc [2]. Understanding the structure, catalytic pathway, and molecular regulation of KGDHc is important because dysfunction in the enzyme complex is associated with the onset of several metabolic and neurological diseases and cancer. Defects in KGDH activity have been associated with the pathogenesis of neurological and metabolic disorders [5,11,12,13,14,15,16,17]. Over-expression of KGDHc occurs in several cancer types, has been linked to metastasis, and defects in the activity of the enzyme complex are associated with the accumulation of 2-hydroxyglutarate, an important oncometabolite [18,19].

## 2. KGDHc Is a Potent mtO_2_^•−^/mtH_2_O_2_ Source

There has been a renaissance in the study of the redox regulatory properties of KGDHc because it has been found to be crucial in mediating intra- and intercellular signals. Here, I will only address the redox signaling functions of KGDHc, but its other roles, such as modulating the cell succinylome and its nuclear localization for epigenetic regulation, have been discussed elsewhere [10]. The main source of mtO_2_^•−^/mtH_2_O_2_ in KGDHc is the FAD group in the E3 subunit (Figure 1). The mtO_2_^•−^/mtH_2_O_2_ generating capacity of KGDHc was first identified in 1969 when it was found that the purified E3 subunit generates O_2_^•−^ through a reverse electron transfer (RET) reaction from NADH [20,21]. This ROS production occurs at the E3 subunit, and the production of O_2_^•−^ and H_2_O_2_ depends on the redox state and chemistry of the FAD and its accessibility to molecular oxygen (O_2_) [22]. Later research confirmed recombinant KGDHc expressed in bacteria or purified from porcine heart can form a mixture of O_2_^•−^/H_2_O_2_ from electron side reactions [23,24,25,26]. Renewed interest in the study of the electron side reactions in KGDH that promote O_2_^•−^/H_2_O_2_ production revealed it is a potent source of mitochondrial ROS (mtROS) in synaptosomes and neurons and skeletal muscle, cardiac, and liver mitochondria [3,4,5,6,7,23,24,27,28,29,30,31,32,33]. This led to the formulation of the hypothesis that KGDHc could cause cellular oxidative distress through the hyper-generation of mtO_2_^•−^/mtH_2_O_2_, contributing to the progression of neurological disorders [7,19,24,34,35]. In this case, it was postulated that the conditions that disable complex I activity in the ETC and promote metabolic gridlock in mitochondria, causing NADH or succinyl-CoA accumulation, can result in the over-reduction of KGDHc, driving up mtROS production. This increased mtROS generation by the KGDHc was proposed to occur through reverse electron transfer (RET) to the E3 subunit from NADH [24,26,36]. Horvath et al. later revealed that this RET-driven mtROS hyper-production by KGDHc, caused by NADH accumulation, may play a significant role in neurological disorders using murine models deficient for for the genes encoding DLST (E2) or DLD (E3) [28]. Site-specific inhibitors for KGDHc, like 2-keto-3-methylvaleric acid (KMV), valproic acid, carboxyethyl succinyl phosphonate (CESP), and the membrane-penetrating compound triethyl succinyl phosphonate (TESP), have been valuable in estimating the rate of mtO_2_^•−^/mtH_2_O_2_ by KGDHc and its potential contribution to oxidative distress [6,30,37,38] (Figure 1). Using these tools, it has been shown that KGDHc can be a much more potent mtO_2_^•−^/mtH_2_O_2_ source when compared to complex I and displays a rate of mtROS generation that is close to complex III [6,32]. Recent identification of the S1QEL (S1) and S3QEL (S3) compounds, which are high-affinity and site-specific inhibitors for electron leaks from complexes I and III, respectively, that also do not alter OxPhos, has also been highly advantageous in the study of KGDHc-mediated mtO_2_^•−^/mtH_2_O_2_ production [39,40,41]. This is because previous studies that investigated KGDHc production in isolated mitochondria also used classic ETC inhibitors that alter mitochondrial electron fluxes, which changes the rate of mtO_2_^•−^/mtH_2_O_2_ genesis from sources upstream of complex I and complex III. Using S1 and S3 compounds in combination with KGDHc inhibitors like KMV and VA, it was recently shown that KGDHc is a more potent mtO_2_^•−^/mtH_2_O_2_ source in liver mitochondria when compared to the ETC [30,41]. Also, this combined approach led to the discovery that fatty acid oxidation in liver mitochondria produces most of its mtO_2_^•−^/mtH_2_O_2_ through KGDHc and not the ETC, which could have strong implications for understanding lipotoxicity and the manifestation of disorders caused by aberrant intracellular lipid accumulation like NAFLD and its progression to more serious hepatic diseases like non-alcoholic steatohepatitis (NASH), cirrhosis, and hepatocellular carcinoma (HCC) [30,41,42].

Deploying selective blockers for KGDHc could be exploited to prevent mtO_2_^•−^/mtH_2_O_2_ hyper-generation for the treatment of metabolic and neurological diseases, inflammation, and cancer. As discussed above, dietary fat overload may trigger the manifestation of NAFLD through the hyper-generation of mtO_2_^•−^/mtH_2_O_2_ by KGDHc. In a recent study, data were collected showing that subjecting male C57BL6N mice to a high-fat diet for only 6 weeks resulted in the manifestation of NAFLD caused by oxidative distress triggered through KGDHc mtO_2_^•−^/mtH_2_O_2_ hyper-generation, which could be suppressed by KMV [43]. Thus, the early manifestation of NAFLD may be prevented by selectively interfering with mtO_2_^•−^/mtH_2_O_2_ by KGDHc. Phosphonates like succinyl phosphonates (SP) and their triethyl, phosphoethyl, and carboxyethyl esters have been shown to be highly effective inhibitors of KGDHc activity and thus strongly interfere with mtO_2_^•−^/mtH_2_O_2_ genesis [37,38]. These compounds have been successfully used to mitigate glutamate excitotoxicity and anxiety in rats through short and temporary inhibition of KGDHc [44,45]. Additionally, CESP protects cultured neuroblastoma cells from oxidative distress through the short-term inhibition of KGDHc [46]. Lipoate analog devimistat (CPI613) has also been shown to be an effective inhibitor of mtO_2_^•−^/mtH_2_O_2_ production by KGDHc [33,47]. Notably, devimistat is under clinical trial and is effective in the killing of cancer cells by disrupting redox homeodynamics [19]. Several studies have been conducted showing that the administration of devimistat alone or in combination with other chemotherapeutics is highly effective in the eradication of several cancer cell types [48,49,50,51]. This is highly significant because KGDHc has been shown to display increased expression and activity in several cancer types, including metastatic ones [18,52]. It cannot be ignored that in many cancer cases, KGDHc is a tumor suppressor because its decreased activity promotes HIF-1α stabilization and 2-hydroxyglutarate accumulation, factors that promote tumorigenesis and increase metastatic potential [53,54]. However, the fact that mtROS over-production also promotes oncogenesis and metastasis implicates KGDHc over-expression in the progression of some cancers as well.

## 3. Dynamic Redox Control of KGDHc May Be a New Therapeutic Approach to Prevent Oxidative Distress

Many investigations aimed at interrogating the effect of oxidative distress on mitochondria have shown that bioenergetics is reversibly regulated by changes in cell redox homeodynamics (reviewed in [3,55,56,57,58,59]). For example, supplying isolated mitochondria with a bolus of H_2_O_2_ strongly inhibits the ETC and OxPhos, which can be recovered once the H_2_O_2_ is consumed by antioxidant defenses [60]. This reversible regulation of mitochondrial bioenergetics occurs through the oxidation and reduction of modifiable protein cysteine thiols (Figure 2). It is now well known that there are many mitochondrial proteins (and others throughout the cell) that can be site-specifically and reversibly regulated through covalent redox modifications like S-glutathionylation, C-glutathionylation, S-nitrosation, S-sulfoxidation, S-cysteinylation, S-homocysteinylation, S-CoAlation, and many more (Figure 2) [61]. In fact, the Oximouse model estimated there are approximately 34,000 cysteines in 9400 proteins that can be reversibly modified by a redox modification [62]. Moreover, it has been predicted that oxidizable cysteine thiols can adopt several different redox-modified forms, or “oxiforms”, based on the type of redox modification that occurs [63,64]. Proteins with more than one oxidizable cysteine can adopt many functionally distinct oxiforms [63,64]. Together with the many types of oxidants that modify cysteines, cells could adopt distinct “redox signatures” based on protein oxiform heterogeneity, which likely plays a significant role in dictating the outcome of adaptive cell signals [63,64]. Mitochondria contain many of these reactive cysteines because the organelle is rich in proteinaceous sulfur and harbors a redox environment conducive to driving these reactions [56,58]. These redox reactions are critical for modulating ETC and OxPhos function, TCA cycle flux, proton leaks and solute transport, mitochondrial fission and fusion, protein import, mitochondria-to-cell redox signaling, ion fluxes, and the prevention of oxidative distress [56,58,65,66]. Crucially, disruption of these redox regulatory pathways in mitochondria is associated with the onset of multiple diseases, the progression of cancer, and inflammation. This is related to the hyper-generation of mtO_2_^•−^/mtH_2_O_2_ and the prolonged oxidation of antioxidant defenses, which results in the non-specific and irreversible oxidation of protein cysteine thiols, which disables mitochondrial functions (e.g., otherwise now called oxidative distress). The targeting of mitochondria-selective antioxidants to the matrix has been shown to prevent this oxidative distress by not only mitigating the over-generation of mtO_2_^•−^/mtH_2_O_2_ but also restoring protein redox regulation. For example, it has been shown that mitochondria-selective antioxidant elamipretide (SS-31) protects against age-related sarcopenia and improves exercise tolerance in aged mice by restoring matrix redox homeodynamics and preventing the deactivation of OxPhos caused by the non-selective over-glutathionylation of mitochondrial proteins [67,68]. The elamipretide treatment also preserves cardiac function in aged mice, which is due to the prevention of glutathione pool oxidation and dysfunctional S-glutathionylation reactions [69,70,71]. Disruption of cell redox homeodynamics has also been found to play a role in cancer development and metastasis. For instance, defects in the protein S-glutathionylation of tumor suppressor proteins, glycolytic enzymes, the epigenome, and mitochondrial proteins have been found to promote tumorigenesis and increased resistance towards chemotherapy (reviewed in [72]). Recent work has demonstrated that over-expression of glutaredoxin-2 (Glrx2), the thiol oxidoreductase that mediates reversible S-glutathionylation reactions in mitochondria, is a prognostic factor for the survival of patients with colorectal cancer [73]. Chemical induction of S-glutathionylation of uncoupling protein-2 (UCP2), which is over-expressed in several cancer types to confer resistance to oxidative distress, sensitizes drug-resistant promyelocytic leukemia cells to chemotherapeutic agents [74]. Thus, targeting the redox-sensing properties of KGDHc could have therapeutic potential for the treatment of some cancers.

### 3.1. Reversible S-Glutathionylation and S-Nitrosation Regulate mO_2_^•−^/mH_2_O_2_ Production by KGDHc

KGDHc is highly susceptible to deactivation through the cysteine oxidation of its vicinal lipoic acid thiols on the E2 subunit (Figure 2) [43,47,75,76,77,78]. S-sulfoxidation, more specifically the oxidation of the vicinal thiol to corresponding sulfinic (SO_2_H) and sulfonic acids (SO_3_H) or formation of a covalent adduct with lipid peroxidation end products like 4-hydroxy-2-nonenal (4-HNE) irreversibly deactivate KGDHc (Figure 2) [4,5,75]. This occurs during oxidative distress and correlates with neurodegeneration. However, other redox modifications to KGDHc, like S-glutathionylation and S-nitrosation, are reversible and protect the E2 subunit from the formation of irreversible adducts [76,77,79]. Notably, several studies have also shown that the S-glutathionylation or S-nitrosation of KGDHc on its E2 subunit abrogates mtO_2_^•−^/mtH_2_O_2_ production by the E3 subunit [43,47,78]. The S-glutathionylation of KGDHc is a self-contained negative feedback loop that mitigates oxidative distress by nullifying the over-production of mtO_2_^•−^/mtH_2_O_2_ (Figure 2). In this mechanism, an uptick in mtH_2_O_2_ levels results in reduced glutathione (GSH) oxidation, driving the S-glutathionylation of KGDHc (Figure 2). This decreases mtO_2_^•−^/mtH_2_O_2_ genesis by KGDHc and any other mtROS sources that are downstream in the ETC (Figure 2). The subsequent downtick in mtO_2_^•−^/mtH_2_O_2_ production promotes recovery of mitochondrial redox buffering capacity through the NADPH-driven reduction of glutathione disulfide (GSSG) (Figure 2). This, in turn, leads to the Glrx2-mediated deglutathionylation of KGDHc and the reactivation of the enzyme complex. Through this mechanism, mitochondria can control the strength and duration of their mtO_2_^•−^/mtH_2_O_2_ capacity, which is critical for preventing oxidative distress and modulating mitochondria-to-cell redox signals. KGDHc is also targeted for reversible S-nitrosation (Figure 2) [47,79,80,81,82]. Experimental evidence has shown that the S-glutathionylation of the E2 subunit of KGDHc occurs on one of the two vicinal thiol groups [76]. This means the second vicinal thiol could be available to resolve the S-glutathionylation for reactivation of the KGDHc enzyme, although it has been found that Glrx2 (and Glrx1 in in vitro experiments) are needed to induce E2 deglutathionylation [3,10]. As described above, it is predicted that proteins can adopt many oxiforms, which is based on the number of redox-sensitive cysteines in the protein and the type of redox modification that occurs on the sulfur [63,64]. In addition, glutathione S-transferase P (GST-P), thioredoxins, and peroxiredoxins have also been shown to mediate S-glutathionylation reactions [83,84,85]. Evidence has also been generated showing KGDHc and PDHc can be S-glutathionylated on the E1 subunit, which affects mtROS generation [25,86]. Together, although it has been shown KGDHc is modulated through the S-glutathionylation of the lipoate in the E2 subunit, it is also feasible that KGDHc (1) can be S-glutathionylated by GST-P, thioredoxin, or peroxiredoxin and (2) may adopt multiple oxiforms based on the type of sulfur oxidation or whether it occurs on the E1 or E2 subunits, which could play a fundamental role in the modulation of mtROS production. The ability of KGDHc to adopt different oxiforms and be targeted by other oxidoreductases like thioredoxins is reflected by the fact that it undergoes S-nitrosation. The S-nitrosation of KGDHc is required to transiently inhibit the TCA cycle in macrophages and inhibit mtROS production in liver cells. This modulates the generation of immunomodulatory metabolites like itaconate, succinate, and 2-hydroxyglutarate and the regulation of the inflammatory response [80,81]. It is unknown how S-nitrosation is reversed, but it is thought to be facilitated by thioredoxins [87]. Nonetheless, S-nitrosation, either with S-nitroso-glutathione or other nitro-donating species, like MitoSNO, can preserve mitochondrial bioenergetics and be protective against several diseases [88,89,90,91].

### 3.2. The Redox Sensing Properties of KGDHc Could Be a Therapeutic Target for the Treatment of NAFLD

NAFLD occurs in 25% of the population in North America, and its surge is associated with the prevalence of obesity, type 2 diabetes mellitus (T2DM), and metabolic syndrome [92,93]. The manifestation of NAFLD begins with simple steatosis, which is characterized by the accumulation of intracellular lipid deposits in ~5% of hepatocytes [94,95]. This can be accompanied by the over-production of ROS by mitochondria, but without any significant tissue damage or inflammation [96,97]. NAFLD manifestation is considered benign, but if left untreated, in some cases, can progress to non-alcoholic steatohepatitis (NASH), which is characterized by inflammation, hepatocellular ballooning and metabolic dysfunction, cell damage caused by oxidative distress, and fibrosis [92,98,99]. Persistent inflammation and hepatocellular damage trigger hepatic stellate cells, resulting in increased fibrosis, leading to cirrhosis and eventually hepatocellular carcinoma [100]. Importantly, the onset of simple steatosis and its transition to more serious liver diseases are influenced by several genetic and environmental factors, and therefore its progression is nonlinear and complex. Despite the surge in fatty liver diseases, effective curative approaches for the treatment of NAFLD have been limited because cases are asymptomatic and thus its manifestation is difficult to detect until the onset of cirrhosis [101,102]. Dietary habits play a critical role in the onset of simple steatosis and its progression to NASH. It was recently established that chronic overconsumption of highly processed foods rich in saturated fats and refined fructose-containing sugars significantly contributes to the onset of NAFLD [103,104,105]. This has been correlated with the induction of hepatocellular oxidative distress and mitochondrial dysfunction [43,96,97,103,104,105]. In addition, NAFLD occurs more frequently in men compared to fertile women [106]. Menopause can significantly increase the risk for the development of NAFLD in women, which parallels the onset of other metabolic disorders like obesity and T2DM [93].

Mitochondria can account for up to 90% of the total ROS formed in mammalian cells and are therefore significant sources of hepatocellular oxidative distress. Thus, mitochondria-targeted therapies that abrogate the hyper-production of mtO_2_^•−^/mtH_2_O_2_ caused by nutrient overload hold significant promise for preventing/reversing NAFLD and NASH. Mitochondria-targeted compounds that scavenge various ROS, like MitoQ, Mito-Vitamin E, MitoTEMPO, and AntiOxCIN_4_, have been shown to abrogate NAFLD caused by nutrient overload [107,108,109]. Recent work by our group has sought to target the redox sensor properties of KGDHc to mitigate mtO_2_^•−^/mtH_2_O_2_ hyper-production caused by nutrient overload in certain diseases. Using a mouse model ablated for the *Glrx2* gene (*Glrx2^−/−^*), we were able to show that the induction of KGDH S-glutathionylation protects male C57Bl6N mice from the manifestation of non-alcoholic fatty liver disease (NAFLD) caused by dietary fat overload (Figure 3) [43]. This protective effect was due to the S-glutathionylation-mediated inhibition of mtO_2_^•−^/mtH_2_O_2_ hyper-production by KGDHc, which prevented oxidative distress caused by the high-fat diet (Figure 3) [43]. Lipotoxicity caused by dietary fat overload in wild-type male mice resulted in GSSG accumulation, induction of oxidative distress, and the aberrant increase in non-specific protein S-glutathionylation of mitochondrial proteins [43]. This correlated with micro- and macro-vascularization of liver parenchymal cells, intrahepatic lipid accretion, and the onset of fibrosis, hallmarks for NAFLD and the development of NASH [43]. Ablating the *Glrx2* gene in the male mice fed the high-fat diet negated the accumulation of GSSG and promoted the biosynthesis of GSH and an increase in overall mitochondrial redox buffering capacity, which is likely the reason why the regulatory function of reversible S-glutathionylation was restored [43]. It is crucial to point out that Glrx2 S-glutathionylates and deglutathionylates target proteins in mitochondria, which, as described above and in Figure 2, occurs in response to the redox poise of the GSH pool. Therefore, it seems counterintuitive that ablating the *Glrx2* gene is protective against NAFLD development. However, as discussed above, reversible S-glutathionylation reactions can also be mediated by GST-P, thioredoxins, and peroxiredoxins, which may compensate for the loss of Glrx2. In addition, Grayson et al. provided evidence showing ablation of *Glrx2* increases GSH levels and protects from oxidative distress [43]. Therefore, the beneficial effects of deleting *Glrx2* are not only related to the inhibition of mtO_2_^•−^/mtH_2_O_2_ hyper-production by KGDHc, but also through the preservation of mitochondrial redox homeodynamics in response to hepatic nutrient overload. In Grayson et al., we also serendipitously identified succinate, aspartate, and GSSG as sex-dependent circulating biomarkers for NAFLD manifestation that accumulate in the plasma collected from male, but not female, C57BL6N mice (Figure 3) [43]. Indeed, succinate, aspartate, and GSSG all accumulated several-fold in the plasma of wild-type male mice subjected to dietary fat overload (Figure 3) [43]. Succinate has been shown to accumulate in the blood following the onset of NAFLD and obesity in mouse models and humans [110,111,112]. However, what was highly notable in Grayson et al. is that the ablation of the *Glrx2* gene completely mitigated the accumulation of succinate, aspartate, and GSSG in the plasma of the male mice fed the high-fat diet (Figure 3) [43]. These novel findings show that the targeted regulation of the redox properties of KGDH in male mouse models for NAFLD mitigates the onset of this disease in response to dietary fat overload and that the metabolites succinate, aspartate, and GSSG are promising biomarkers that could be used to diagnose the disease in its early stages.

Recently, our team also tested whether mitochondria-targeted S-nitrosating agent, MitoSNO, could also elicit a cytoprotective effect like that observed in the livers of the *Glrx2^−/−^* male mice. MitoSNO is a mitochondria-selective NO^•^ donor that was developed to study the cytoprotective effects of protein S-nitrosation [113]. This selectivity for mitochondria is driven by triphenylphosphonium ion, which promotes the accumulation of MitoSNO in the matrix by several hundred-fold [113]. MitoSNO-mediated nitro-group donation to mitochondrial proteins is effective at preventing ischemia-reperfusion injury to cardiac, brain, and skeletal muscle tissue and post-myocardial infarction heart failure [113,114,115,116,117,118]. This is achieved through the S-nitrosation of Cys^39^ in the ND3 subunit of complex I, which nullifies mtROS hyper-generation in the myocardium, preventing oxidative distress [114]. Applying MitoSNO to liver mitochondria revealed that it blocks mtO_2_^•−^/mtH_2_O_2_ generation by KGDHc through its S-nitrosation (Figure 3) [41]. Importantly, MitoSNO mitigated lipotoxicity caused by fat and fructose overload in cultured Huh-7 human hepatoma cells by limiting mtO_2_^•−^/mtH_2_O_2_ hyper-generation by KGDHc and preventing oxidative distress, cell death, and the accumulation of intracellular lipids [41]. Additionally, cultured Huh-7 cells subjected to fat and fructose overload disabled OxPhos, which was prevented by MitoSNO treatment [41]. MitoSNO also alleviated the over-generation of mtO_2_^•−^/mtH_2_O_2_ by liver mitochondria isolated from mice fed a high-fat diet (HFD), which coincided with the recovery of OxPhos after its deactivation by the dietary fat overload [41]. More work is needed to determine if MitoSNO can mitigate the progression of NAFLD caused by dietary fat overload in vivo and ascertain if the mitochondria-targeted compound can circumvent the accumulation of succinate, aspartate, and GSSG in plasma. However, when taken together, the MitoSNO compound is a promising therapeutic tool that could be used to abrogate the onset of NAFLD through the targeted and dynamic redox regulation of KGDHc.

## 4. Conclusions

Our scope of understanding how cells invoke mtO_2_^•−^/mtH_2_O_2_ to control various intra- and intercellular signaling cascades has expanded dramatically over the past two decades. This is due, in part, to the development of novel and more sensitive and selective quantitative tools that allow for the precise measurement of mtO_2_^•−^/mtH_2_O_2_ production by various mitochondrial sources. It is now understood that mitochondria contain 12 different mtO_2_^•−^/mtH_2_O_2_ sources [119]. Of these potential generators, KGDHc has emerged as a potent source in various tissues and cell types, including hepatocytes, implicating it in facilitating oxidative eustress signals under normal cellular conditions and serving as a source of oxidative distress in the pathogenesis of disease (e.g., when mtO_2_^•−^/mtH_2_O_2_ undergoes hyper-generation). As discussed above, the hyper-generation of mtO_2_^•−^/mtH_2_O_2_ by KGDHc occurs in several experimental models for disease, including fatty liver disease, which manifests as NAFLD in response to dietary fat overload. Notably, the hyper-production of mtO_2_^•−^/mtH_2_O_2_ by KGDHc can be countered by its S-glutathionylation or S-nitrosation, attenuating the onset of NAFLD induced by hepatic nutrient overload. Thus, taking advantage of the redox sensing properties of KGDHc could be used to mitigate the manifestation and progression of metabolic diseases like NAFLD. In addition, the effectiveness of the targeted pharmacological manipulation of KGDHc to abrogate NAFLD manifestation and other diseases using mitochondria therapies could potentially be monitored by measuring plasma levels of succinate, aspartate, and GSSG. Although more investigations are needed, the targeted redox regulation of KGDHc could be a promising pharmacological target for the treatment of NAFLD. In addition, tracking circulating succinate, aspartate, and GSSG (and other metabolites like itaconate) may serve as a new clinical tool to diagnose NAFLD in its early stages, track its progression, and gauge the effectiveness of therapeutics in the prevention or reversal of the disease.

## Figures and Tables

**Figure 1 cells-14-00653-f001:**
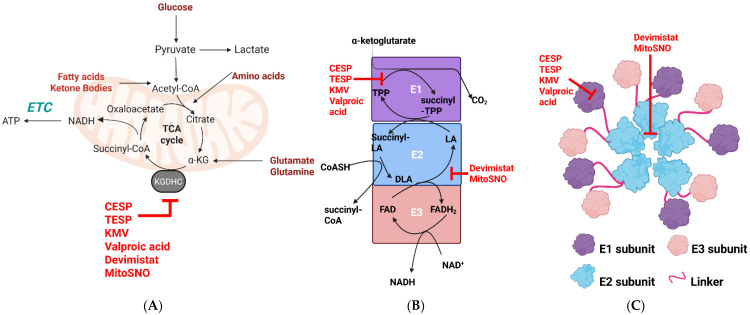
Schematic representation of the structure and catalytic mechanism of KGDHc and its role in mitochondrial metabolism. (**A**) KGDHc intersects monosaccharide, fatty acid, ketone body, and amino acid metabolism in the TCA cycle with the electron transport chain (ETC) and oxidative phosphorylation (OxPhos). KGDHc inhibitors are depicted in red. (**B**) The catalytic mechanism of KGDHc. α-ketoglutarate generated from monosaccharide, fatty acid, or amino acid metabolism is decarboxylated by the E1 (α-ketoglutarate decarboxylase) subunit, resulting in the formation of succinyl-TPP. The E2 subunit (dihydrolipoamide succinyltransferase; DLST) transfers the succinyl group to CoA, forming succinyl-CoA. This generates dihydrolipoamide, which is oxidized by the FAD center through a CxxxxC redox motif in the E3 subunit (dihydrolipoamide dehydrogenase; DLD). The FADH_2_ is used for NADH biosynthesis. Electrons leak from the E3 through side reactions that produce mitochondrial superoxide (mtO_2_^•−^) and/or mitochondrial hydrogen peroxide (mtH_2_O_2_). Inhibition sites for CESP, TESP, KMV, valproic acid, MitoSNO, and devimistat are depicted in red. (**C**) Proposed structural model for KGDHc. The image was reproduced from Zhang et al. [2]. Details on how Zhang et al. solved the molecular architecture of KGDHc can be found in [2]. Sites of inhibition of the individual subunits by CESP, TESP, KMV, valproic acid, MitoSNO, and devimistat. The image was generated using Biorender.

**Figure 2 cells-14-00653-f002:**
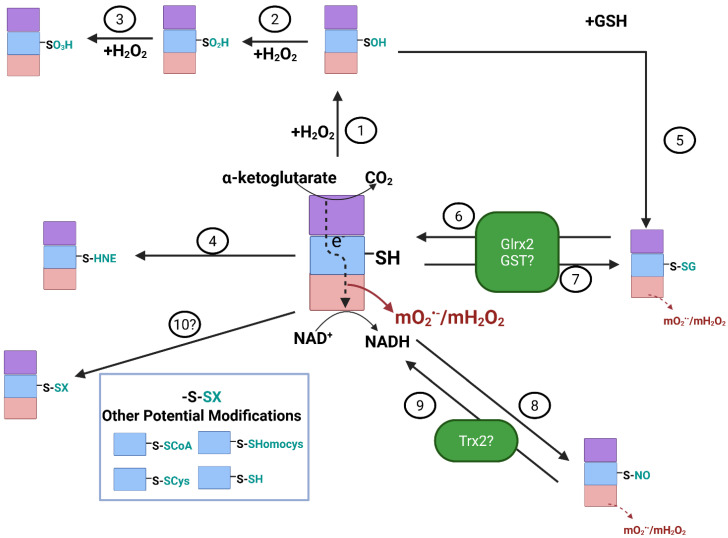
Reversible redox regulation of E2 subunit of KGDH is required to control mtO_2_^•−^/ mtH_2_O_2_ production by the E3 subunit. H_2_O_2_ can induce the sulfenylation (-SOH) of the vicinal lipoate thiols in the E2 subunit (1). Conditions that promote the over-production and accumulation of H_2_O_2_ can cause the oxidation of the sulfenic acid to a sulfinic (-SO_2_H) (2) and sulfonic (-SO_3_H) (3) acid. Note that the sulfinic acid can be reduced by sulfiredoxin. However, formation of a sulfonic acid is irreversible. The vicinal lipoate thiol can also be modified by aldehydic lipid peroxidation end products like 4-hydroxy-2-nonenal (4-HNE) (4), forming an irreversible Michael adduct with KGDHc. Further oxidation of -SOH to -SO_2_H can be prevented by S-glutathionylation (5), which occurs through the spontaneous or enzymatic addition of GSH to the oxidized thiol. Protein S-glutathionylation of the vicinal lipoate thiol can also occur directly through glutaredoxin-2 (Glrx2) (6), a reaction that first requires oxidation of GSH to GSSG by glutathione peroxidases (GPx). Glrx2 enzymes are small heat-stable thiol oxidoreductases that catalyze reversible thiol disulfide exchange reactions based on the oxidation state of the GSH pool. Thus, reduction of GSSG back to GSH by NADPH-dependent glutathione reductases (GR) induces the Glrx2-mediated deglutathionylation of KGDH (6). Oxidation of GSH also activates the glutathionyltransferase activity of Glrx2, resulting in KGDHc S-glutathionylation (7). Glutathione S-transferases have also been speculated to carry out the S-glutathionylation of KGDHc (7). The vicinal thiols of the E2 subunit of KGDH can also undergo S-nitrosation, which has been shown to be induced by S-nitroso-glutathione (GSNO) or mitochondria-targeted S-nitrosation agent (MitoSNO) (8). Although it is not known how KGDHc is denitrosated, it is likely to be driven by thioredoxin-2 (Trx2) or other antioxidant defense enzymes (9). Note that reactions 6-9 play an important role in modulating mtO_2_^•−^/mtH_2_O_2_ production by KGDH. This is because S-glutathionylation or S-nitrosation blocks electron flow from the E2 subunit to the mtO_2_^•−^/mtH_2_O_2_ producing E3 subunit. Finally, although speculative, it is possible KGDH may be targeted by other redox reactions such as S-CoAlation, S-cysteinylation, S-homocysteinylation, and S-persulfidation (10). Image was generated using Biorender.

**Figure 3 cells-14-00653-f003:**
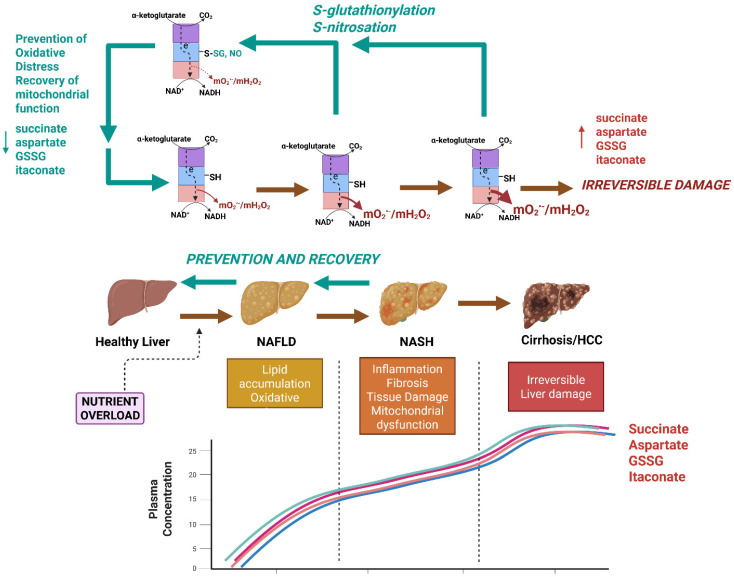
Nutrient overload caused by the chronic overconsumption of calorie-dense foods (e.g., Western-style diet) triggers the onset of simple steatosis and non-alcoholic fatty liver disease (NAFLD) through the induction of the hyper-production of mtO_2_^•−^/mtH_2_O_2_ by KGDH. Nutrient overload caused by chronic exposure to a diet rich in processed sugars and saturated fats causes simple steatosis, which is characterized by intrahepatic lipid accumulation and oxidative stress. If left untreated, the NAFLD can progress to more serious forms of the disease like non-alcoholic steatohepatitis (NASH), which is characterized by inflammation, fibrosis, tissue damage, and hepatic ballooning. NASH can then cause cirrhosis and hepatocellular carcinoma (HCC). Succinate, aspartate glutathione disulfide (GSSG), and itaconate accumulate in the bloodstream after induction of NAFLD and thus may be sex-dependent biomarkers for the early development of the disease. NAFLD and NASH are reversible, and treatment plans could be implemented after the detection of the disease using succinate, aspartate, GSSG, and itaconate. Prevention or reversal of NAFLD and NASH could be achieved through the targeted redox modification of KGDH. KGDH has been identified as a potent mtO_2_^•−^/mtH_2_O_2_ source in the livers of mice subjected to dietary fat overload. S-glutathionylation or S-nitrosation (induced by MitoSNO) mitigates mtO_2_^•−^/mtH_2_O_2_ hyper-production, limiting oxidative distress and tissue damage, which restores hepatic health. The effectiveness of the dynamic regulation of KGDHc using redox modifications for the prevention or treatment of NAFLD can be tracked using succinate, aspartate, GSSG, and itaconate. Red arrows: progression of liver disease caused by hyper-production of mtROS by KGDHc. Green arrows: recovery of liver function through dynamic regulation of KGDHc activity. This image was generated using Biorender.

## Data Availability

Not applicable.

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
