# Peer review of "Targeted Redox Regulation α-Ketoglutarate Dehydrogenase Complex for the Treatment of Human Diseases"

_cells, 2025, doi:10.3390/cells14090653_

Round 1
Reviewer 1 Report
Comments and Suggestions for Authors
The article provides a detailed and well-structured review of the α-ketoglutarate Dehydrogenase 2 Complex (KGDHc), its role in mitochondrial metabolism, and its implications for redox regulation and disease like non-alcoholic fatty liver disease. The author critically analyzes current literature, and well-prepared Figures help readers follow the text. I only have one comment regarding more information on a link of KGDHc with cancer progression, because this issue is not clearly described in the review.
Author Response
Response: Thank you for the comment. I have expanded the discussion of these concepts in cancer on pages 6 and 8 in the revised document.
Reviewer 2 Report
Comments and Suggestions for Authors
This review is an excellent contribution updating information that has emerged regarding KGDHc and its contributions to mito superoxide/H2O2 production, particularly in disease states including neurological and metabolic diseases. I am enthusiastic about the important points made and the new literature that is highlighted. I have some comments below, however, because there are aspects that are confusing me a bit. I bring these points up here as they may highlight places where some extra things could be added for clarification.
There is a puzzle regarding glutathionylation of lipoic acid to me, which is that if just one GSH forms a -SSG modification on one of the sulfurs, I would expect that the other (vicinal) thiol would be highly likely to come in and resolve that to oxidized lipoate, with GSH being released. What might prevent that would be if high levels of oxidant or GSSG would cause both sulfurs to be modified at the same time, or one of the sulfurs is otherwise blocked, e.g. by an electrophile. This leads me to note that only a single thiol group is shown on E2 in Fig. 2. This is good for illustrating in a simple way the chemistry involved. But this then looks more like what would happen at a single cysteine residue rather taking into account the other sulfur of the oxidized lipoamide ring. I’m not making any specific recommendation here, but it would be helpful to add something more about this.
Another puzzle I have is about how blocking KGDHc activity dials down mtH2O2 generation by E3. Certainly normal catalysis does supply electrons to E3, and the FADH2 can then reduce oxygen instead of NAD+. But also high NADH would cause the same thing, and I can imagine that not being particularly dependent on the catalytic activity of E2. This is not shown in figures nor dealt with in the text it seems. Isn’t NADH accumulation a big part of mtH2O2/O2-production, including in these enzymes?
There is also still a puzzle to me that removing Grx2 is a good thing and lowers mtH2O2 levels. It is pretty much said already that this protein catalyzes both addition and removal of glutathione to/from thiols (or at least cysteine thiols, not sure if this is actually known for lipoyl-SSG), with the direction of the reaction depending on relative GSH and GSSG levels (thus, it usually reduces glutathionylated sites). I’m not sure that there is any change to be made unless there is a way to clarify more about how/why removal of Grx2 would be helpful to the cells as it seems like this would only slow down reactions that would occur non-enzymatically (or maybe it is related to the specificity that can be imparted when a protein is involved in the reaction).
Line 219, it looks to me like reference 24 found the opposite of what is stated here regarding glutathionylation of E2 and mtH2O2 production.
Line 236, “or” should come before “nitrite”, but actually I don’t think “nitrite” is correct here (at least not as a direct agent), so perhaps substitute in “other nitro donating species” or something like that.
Small typos/corrections
I recommend referring briefly to the disulfide/dithiol redox center also in E3 that mediates the electron transfer between the dithio-lipoate and FAD, as it is often left out of schemes. I suggest leaving the scheme simple as shown but adding a short mention in the legend. Line 57-58 could be something like “which is oxidized by the FAD center in the E3 subunit mediated by a CxxxxC dithiol/disulfide redox center”
Line 45 at the end seems to sound better as “transfers a hydride to NAD+” (rather than “hydrides”)
Line 78, insert “and”, “and has been linked”. There is another “and” after the comma, but that is not a similar/parallel phrase.
Line 127, I think it reads better by inserting “that”, “showing that subjecting male…”
Line 147, change “on” to “in”
Line 267, delete “the”
Line 329, “subjected” should be moved to after “hepatocytes”
Line 344 mentions 12 different mito superoxide/H2O2 sources but doesn’t list them; I would at least add a specific reference for the reader here.
Lines 353-356 sentence seems it needs rephrasing, “could be used” doesn’t read quite right.
Author Response
There is a puzzle regarding glutathionylation of lipoic acid to me, which is that if just one GSH forms a -SSG modification on one of the sulfurs, I would expect that the other (vicinal) thiol would be highly likely to come in and resolve that to oxidized lipoate, with GSH being released. What might prevent that would be if high levels of oxidant or GSSG would cause both sulfurs to be modified at the same time, or one of the sulfurs is otherwise blocked, e.g. by an electrophile. This leads me to note that only a single thiol group is shown on E2 in Fig. 2. This is good for illustrating in a simple way the chemistry involved. But this then looks more like what would happen at a single cysteine residue rather taking into account the other sulfur of the oxidized lipoamide ring. I’m not making any specific recommendation here, but it would be helpful to add something more about this.
Response: This has been reviewed by my group and others in several papers, which is the reason why I chose not to add a discussion on the chemistry of the S-glutathionylation of the lipoate sulfurs in the original submission (1-3). In light of this reviewer’s comment, I understand the importance of adding some information on this topic. A short discussion of the redox modifications that could occur on the lipoate and how KGDHc could adopt different oxiforms has been added to page 9 and 10 of the revised article.
Another puzzle I have is about how blocking KGDHc activity dials down mtH2O2 generation by E3. Certainly normal catalysis does supply electrons to E3, and the FADH2 can then reduce oxygen instead of NAD+. But also high NADH would cause the same thing, and I can imagine that not being particularly dependent on the catalytic activity of E2. This is not shown in figures nor dealt with in the text it seems. Isn’t NADH accumulation a big part of mtH2O2/O2-production, including in these enzymes?
Response: Yes, NADH accumulation due to defects in complex I activity or decreased ETC function can increase mtROS production by KGDHc (and other alpha-keto acid dehydrogenases). This has been reviewed and discussed elsewhere (1, 4), but I have added information on this on page 5 of the revision.
There is also still a puzzle to me that removing Grx2 is a good thing and lowers mtH2O2 levels. It is pretty much said already that this protein catalyzes both addition and removal of glutathione to/from thiols (or at least cysteine thiols, not sure if this is actually known for lipoyl-SSG), with the direction of the reaction depending on relative GSH and GSSG levels (thus, it usually reduces glutathionylated sites). I’m not sure that there is any change to be made unless there is a way to clarify more about how/why removal of Grx2 would be helpful to the cells as it seems like this would only slow down reactions that would occur non-enzymatically (or maybe it is related to the specificity that can be imparted when a protein is involved in the reaction).
Response: This is a good point. I have expanded on this on two pages: page 9, wherein I discuss the other enzymes that could modify KGDHc and how the complex may adopt different oxiforms and again on page 12 where these concepts are discussed in the context of the findings demonstrating ablating Glrx2 protects from NAFLD.
Line 219, it looks to me like reference 24 found the opposite of what is stated here regarding glutathionylation of E2 and mtH2O2 production.
Response: corrected.
Line 236, “or” should come before “nitrite”, but actually I don’t think “nitrite” is correct here (at least not as a direct agent), so perhaps substitute in “other nitro donating species” or something like that.
Response: corrected.
I recommend referring briefly to the disulfide/dithiol redox center also in E3 that mediates the electron transfer between the dithio-lipoate and FAD, as it is often left out of schemes. I suggest leaving the scheme simple as shown but adding a short mention in the legend. Line 57-58 could be something like “which is oxidized by the FAD center in the E3 subunit mediated by a CxxxxC dithiol/disulfide redox center”
Response: Thank you for the recommendation. Corrected.
Line 45 at the end seems to sound better as “transfers a hydride to NAD+” (rather than “hydrides”)
Response: corrected.
Line 78, insert “and”, “and has been linked”. There is another “and” after the comma, but that is not a similar/parallel phrase.
Response: corrected.
Line 127, I think it reads better by inserting “that”, “showing that subjecting male…”
Response: corrected.
Line 147, change “on” to “in”
Response: corrected.
Line 267, delete “the”
Response: corrected.
Line 329, “subjected” should be moved to after “hepatocytes”
Response: corrected.
Line 344 mentions 12 different mito superoxide/H2O2 sources but doesn’t list them; I would at least add a specific reference for the reader here.
Response: corrected.
Lines 353-356 sentence seems it needs rephrasing, “could be used” doesn’t read quite right.
Response: corrected.
- Mailloux, R. J. (2024) The emerging importance of the alpha-keto acid dehydrogenase complexes in serving as intracellular and intercellular signaling platforms for the regulation of metabolism Redox Biol 72, 103155 10.1016/j.redox.2024.103155
- McLain, A. L., Szweda, P. A., andSzweda, L. I. (2011) alpha-Ketoglutarate dehydrogenase: a mitochondrial redox sensor Free Radic Res 45, 29-36 10.3109/10715762.2010.534163
- McLain, A. L., Cormier, P. J., Kinter, M., andSzweda, L. I. (2013) Glutathionylation of alpha-ketoglutarate dehydrogenase: the chemical nature and relative susceptibility of the cofactor lipoic acid to modification Free Radic Biol Med 61, 161-169 10.1016/j.freeradbiomed.2013.03.020
- Tretter, L., andAdam-Vizi, V. (2004) Generation of reactive oxygen species in the reaction catalyzed by alpha-ketoglutarate dehydrogenase J Neurosci 24, 7771-7778 10.1523/JNEUROSCI.1842-04.2004
Reviewer 3 Report
Comments and Suggestions for Authors
Ryan J Malloux’s name is well known among researcher’s working on the field of mitochondria and ROS homeostasis. The present review is an excellent contribution to the field of mitochondrial redox regulation and deals with the role of a-KGDHc in normal and pathological conditions.
The role of oxidative stress in the pathogenesis of various diseases has been widely investigated in the 2000’s, but the antioxidant therapies did not result the expected clinical breakthrough. This review discusses the redox sensing properties of a-KGDHc a promising pharmacological target for the treatment of NAFLD.
Author gives a comprehensive overview about the general importance of a-KGDHc in metabolism under physiological and pathological conditions, and then discusses the role of a-KGDHc in the mitochondrial ROS homeostasis. The great advantage of this subchapter is the brief but accurate description of mitochondrial inhibitors which could serve as useful tools to dissect the role of individual mitochondrial functional units in ROS homeostasis. The site specific reversible covalent redox modifications can modulate the function of mitochondrial electron transport system and therefore the ATP production.
Author finally discusses the implications of the above mentioned mechanisms in understanding the pathogenesis and possible treatments of the non-alcoholic fatty liver disease (NAFLD). Author and coworkers has a valuable contribution on this particular field.
To summarize: the review is correct, written by an expert of the field. It gives the credit to the original papers, (not a review of recent reviews). Figures are adequate and informative. This MS stimulates thinking and gives a perspective for future research efforts.
Author Response
No comments